# Quantum Spatial Search with Electric Potential: Long-Time Dynamics and Robustness to Noise

**DOI:** 10.3390/e24121778

**Published:** 2022-12-05

**Authors:** Thibault Fredon, Julien Zylberman, Pablo Arnault, Fabrice Debbasch

**Affiliations:** 1Université Paris-Saclay, CNRS, ENS Paris-Saclay, INRIA, Laboratoire Méthodes Formelles, 91190 Gif-sur-Yvette, France; 2Sorbonne Université, Observatoire de Paris, Université PSL, CNRS, LERMA, 75005 Paris, France

**Keywords:** quantum algorithms, quantum walks, quantum spatial search, noise

## Abstract

We present various results on the scheme introduced in a previous work, which is a quantum spatial-search algorithm on a two-dimensional (2D) square spatial grid, realized with a 2D Dirac discrete-time quantum walk (DQW) coupled to a Coulomb electric field centered on the the node to be found. In such a walk, the electric term acts as the oracle of the algorithm, and the free walk (i.e., without electric term) acts as the “diffusion” part, as it is called in Grover’s algorithm. The results are the following. First, we run long time simulations of this electric Dirac DQW, and observe that there is a second localization peak around the node marked by the oracle, reached in a time $O(\sqrt{N})$, where *N* is the number of nodes of the 2D grid, with a localization probability scaling as O(1/lnN). This matches the state-of-the-art 2D-DQW search algorithms before amplitude amplification We then study the effect of adding noise on the Coulomb potential, and observe that the walk, especially the second localization peak, is highly robust to spatial noise, more modestly robust to spatiotemporal noise, and that the first localization peak is even highly robust to spatiotemporal noise.

## 1. Introduction

Discrete-time quantum walks (DQWs) correspond to the one-particle sector of quantum cellular automata [1,2]. They can simulate numerous physical systems, ranging from particles in arbitrary Yang–Mills gauge fields [3] and massless Dirac fermions near black holes [4], to charged quantum fluids [5], see also Refs. [6,7,8,9,10,11,12,13,14,15,16] for other physics-oriented applications.

Moreover, DQWs can be seen as quantum analogs of classical random walks (CRWs) [17], and can be used to build spatial-search algorithms that outperform [18] those built with CRWs. Continuous-time quantum walks can also be used for such a purpose [19]. In three spatial dimensions, DQW-based algorithms [18,19] find the location of a marked node with a constant localization probability (We call “localization probability” the probability to be at the marked node, or nodes if there are several of them.) after $O(\sqrt{N})$ time steps, with *N*, the number of nodes of the three-dimensional grid, and this is exactly the bound reached by Grover’s algorithm [20,21,22,23,24,25]. However, no two-dimensional (2D) QW proposed so far reaches Grover’s lower bound.

The state-of-the-art result using a 2D DQW was obtained by Tulsi in Ref. [26]: Tulsi’s algorithm finds a marked node with a localization probability scaling as O(1/lnN) in $O(\sqrt{N})$ time steps, where *N* is the total number of nodes. To reach a probability independent of *N*, several amplitude amplification time steps have to be performed after the quantum-walk part. These extra time steps are Grover’s algorithm time steps, see Ref. [27]. Taking the amplitude amplification into account, Tulsi’s algorithm reaches an O(1) localization probability after $O(\sqrt{N\ln N})$ time steps.

Other schemes of 2D DQW for spatial search have followed, such as the one by Roget et al. in Ref. [28], where the 2D DQW simulates a massless Dirac fermion on a grid with defects. This scheme is inspired by physics, and it reaches Tulsi’s bound using a coin of dimension 2 instead of 4. Recently, Zylberman and Debbasch introduced in Ref. [29] a new DQW scheme for 2D quantum spatial search. This scheme implements quantum search by simulating the dynamics of a massless Dirac fermion in a Coulomb electric field centered on the nodes to be found. We call this DQW “electric Dirac DQW” (We call “Dirac DQW” a DQW that has as a continuum limit the Dirac equation. Throughout this paper, the terminology “electric Dirac DQW” will always refer to a Dirac DQW coupled to a *Coulomb* electric potential, unless otherwise stated. In the literature, other types of electric potentials have been considered. The reason why we do not specify the term “Coulomb” in the present denomination “electric Dirac DQW” is because the idea we want to convey is that the marked node is encoded in the shape of the electric potential, but the precise form of the electric potential, e.g., here, the fact that it is a Coulomb potential, may not be that relevant.). In this walk, the oracle is a position-dependent *phase*.

This oracle is diagonal in the position basis and can be efficiently implemented on *n* qubits up to an error ϵ using $O(\frac{1}{\epsilon })$ primitive quantum gates [30]. This total number of quantum gates is independant of *n* and makes possible the implementation of the oracle on current Noisy Intermediate Scale Quantum (NISQ) devices and on future universal quantum computers.

Note also that the algorithm proposed in Ref. [29] actually constitues a paradigm change in the construction of search algorithms, because it is based on the physically motivated idea that the position of the marked node can be encoded in the shape of an artificial force field, which acts on the quantum walker.

One of the main results of Zylberman and Debbasch’s paper [29] is a localization probability, which displays a maximum in $O_{N\to \infty }\left(1\right)$ time steps, the localization probability scaling as O(1/N) (a detailed analysis is presented in Section 3). Since it focuses on this result, Ref. [29] does not offer an analysis of the walk at times much larger than O(1). Moreover, practical implementation not only on current NISQ devices, but also on future, circuit-based quantum computers, can only be envisaged if the algorithm is robust to noise (see, for example, Refs. [5,31,32,33,34,35,36,37]); this question is also not addressed in Ref. [29].

The aim of this article is to explore both aspects: long-time dynamics and robustness to noise. The main results are the following. First, the electric Dirac DQW exhibits a second localization peak at a time scaling as $O(\sqrt{N})$ with localization probability scaling as O(1/lnN). This makes this walk state-of-the-art for 2D DQW spatial search before amplitude amplification. Moreover, this second localization peak is highly robust to spatial noise. Finally, the peak is also robust to spatiotemporal noise, but not as much as it is to time-independent spatial noise.

The article is organized as follows. In Section 2, we offer a review of the electric Dirac DQW presented in Ref. [29]. In Section 3, we study in detail the first two maxima of the localization probability. We show that the first maximum, already analyzed in Ref. [29], is actually present up to $N=9\times 10^{6}>10^{6}≃2^{20}$ (have in mind that 20 is the current average number of working qubits on most IBM-Q platforms according to https://quantum-computing.ibm.com/services/resources accessed on 29 November 2022). We also present evidence for the scaling laws characterizing both the first peak and the second, long-time peak, which reaches Tulsi’s state-of-the-art bound. In Section 4, we analyze the ressources one needs to implement the quantum spatial search in terms of qubits and primitive quantum operations. In Section 5, we show that the walk, and in particular the second peak, have a good robustness to spatial oracle noise. We also show that the first peak is robust even to spatiotemporal noise. In Section 6, we propose an analysis of the walker’s probability distributions. These probability distributions show that the spatial noise does not affect the shape of the peaks significantly. The peaks remain extremely high relative to the background, which shows not only good but high robustness of the peaks to spatial oracle noise. The probability distributions also show that the second peak is sharper than the first one.

## 2. Basics

### 2.1. Definition of the 2D Electric Dirac DQW

We consider a 2D square spatial grid with nodes indexed by two integers $\left(p,q\right)\in {\left[\;\left[0,M\right]\;\right]}^{2}$, where M∈N is the number of nodes along one dimension and $N=M^{2}$ is the total number of nodes. The time is also discrete and indexed by a label j∈N. The walker is defined by its quantum *state* $|\Psi_{j}\rangle$ in the Hilbert space $H_{C}⊗H_{P}$, where $H_{C}$, called *coin space*, is the two-dimensional Hilbert space, which corresponds to the internal, coin degree of freedom, and $H_{P}$, called *position space*, corresponds to the spatial degrees of freedom. The *wavefunction* of the state will be denoted as $\Psi_{j,p,q}\equiv {\left(\begin{array}{cc} \psi_{j,p,q}^{L} & \psi_{j,p,q}^{R} \end{array}\right)}^{⊤}$, where ⊤ denotes the transposition. The discrete-time evolution of the walker is defined by the following one-step evolution equation,
(1)$$\Psi_{j+1,p,q}={\left(U\Psi_{j}\right)}_{p,q}\;.$$

The one-step evolution operator, also called *walk operator*, U, is defined by
(2)$$U:=e^{-ie\phi }\;R\left(\theta^{-}\right)\;S_{2}\;R\left(\theta^{+}\right)\;S_{1}\;,$$
where $S_{1,2}$ are standard *shift operators*,
(3a)$$\begin{array}{cc} {\left(S_{1}\Psi \right)}_{p,q}^{L} & :=\psi_{p+1,q}^{L} \end{array}$$
(3b)$$\begin{array}{cc} {\left(S_{1}\Psi \right)}_{p,q}^{R} & :=\psi_{p-1,q}^{R} \end{array}$$
(3c)$$\begin{array}{cc} {\left(S_{2}\Psi \right)}_{p,q}^{L} & :=\psi_{p,q+1}^{L} \end{array}$$
(3d)$$\begin{array}{cc} {\left(S_{2}\Psi \right)}_{p,q}^{R} & :=\psi_{p,q-1}^{R}\;, \end{array}$$

R(θ) is a coin-space rotation, also called *coin operator*, defined by
(4)$$R\left(\theta \right)=\left[\begin{array}{cc} \cos\theta & i\sin\theta \\ i\sin\theta & \cos\theta \end{array}\right],$$
and
(5)$$\theta^{\pm }=\pm \frac{\pi }{4}-\frac{\mu }{2}\;,$$
with μ, some real parameter. A schematic representation of a quantum circuit for U is proposed in Figure 1. More details about the circuit are given in Section 4. A schematized picture of the walk operator is proposed in Figure 2.

The operator $e^{-ie\phi }$ is diagonal in position space, i.e., it acts on $\Psi_{j}$ as
(6)$${\left(e^{-ie\phi }\Psi_{j}\right)}_{p,q}=e^{-ie\phi_{p,q}}\Psi_{j,p,q}\;,$$
with $\phi :\left(p,q\right)\mapsto \phi_{p,q}\in R$ some sequence of the lattice position, and *e*, a parameter that we can call the charge of the walker, see why further down. The sequence ϕ can be called the lattice electric potential for at least two reasons: (i) in the continuum limit (see below, Section 2.2), this sequence indeed becomes, mathematically, an electric potential coupled to the walker, who then obeys the Dirac equation, and (ii) beyond the continuum limit, it has been shown that similar 2D DQWs exhibit an exact lattice U(1) gauge invariance [38] which, in the continuum limit, becomes the standard U(1) gauge invariance of the Dirac equation coupled to an electromagnetic potential.

### 2.2. Continuum Limit

We introduce a spacetime-lattice spacing ϵ, and coordinates $t_{j}:=\epsilon j$, $x_{p}:=\epsilon p$, and $y_{q}:=\epsilon q$ [39,40]. We assume that $\Psi_{j,p,q}$ coincides with the value taken at point $t_{j}$, $x_{p}$, and $y_{q}$ by a function Ψ of the continuous coordinates *t*, *x*, and *y*. We are interested in the dynamics followed by Ψ when ϵ→0. Let us introduce the following continuum quantities,
(7a)$$\begin{array}{cc} m & :=\frac{\mu }{\epsilon } \end{array}$$
(7b)$$\begin{array}{cc} V(x_{p},y_{q}) & :=\frac{\phi_{p,q}}{\epsilon }\;, \end{array}$$
which are, respectively, the mass and electric potential, see why just below.

Expand now Equation (Equation 1) in ϵ around ϵ=0. The walk operator, Equation (Equation 2), has been chosen so that (i) the zeroth-order terms give us Ψ(t,x,y)=Ψ(t,x,y), i.e., the terms cancel each other, and (ii) the first-order terms deliver the well-known Dirac equation coupled to an electric potential *V*. This equation (in natural units where c=1 and ℏ=1) reads:(8)$$i\partial_{t}\Psi =H\Psi \;,$$
where the Dirac Hamiltonian is
(9)$$H:=\alpha^{k}\left(-i\partial_{k}\right)+m\alpha^{0}+eV\;,$$
where summation over k=1,2 is implicitly assumed. The alpha matrices are
(10a)$$\begin{array}{cc} \alpha^{0} & :=\sigma_{x} \end{array}$$
(10b)$$\begin{array}{cc} \alpha^{1} & :=\sigma_{z} \end{array}$$
(10c)$$\begin{array}{cc} \alpha^{2} & :=-\sigma_{y}, \end{array}$$
where the σs are the Pauli matrices. Thus, this DQW, Equation (Equation 1), simulates the (1+2)D Dirac equation coupled to an electric potential, explaining why the “Dirac DQW” is called an electric DQW.

### 2.3. Coulomb Potential

As shown in Equation (Equation 8), the sequence ϕ=ϵV represents in the continuum limit the electric potential to which the walker is coupled. We choose *V* to be a Coulomb potential created by a point particle of charge *Q* at location $\left(\Omega_{x},\Omega_{y}\right)$ on the 2D plane: (11)$$eV\left(x,y\right):=\frac{eQ}{\sqrt{{\left(x-\Omega_{x}\right)}^{2}+{\left(y-\Omega_{y}\right)}^{2}}}.$$

For the sake of simplicity, *e* will be set to −1. As discussed in Ref. [29], one can take, without loss of generality (i) $\left(\Omega_{x},\Omega_{y}\right)=\left(\frac{M}{2}-\frac{1}{2},\frac{M}{2}-\frac{1}{2}\right)$, which is called the *center*, and (ii) ϵ=1. The charge *Q* is set to 0.9, and m=μ=0. Notice that the center is not located on a node of the 2D lattice; it is at equal distance of the four nodes, namely, $\left(\frac{M}{2},\frac{M}{2}\right)$, $\left(\frac{M}{2}-1,\frac{M}{2}\right)$, $\left(\frac{M}{2},\frac{M}{2}-1\right)$, and $\left(\frac{M}{2}-1,\frac{M}{2}-1\right)$. With this choice of potential, the walk can be referred to as a “Coulomb walk”.

### 2.4. Definition of the Spatial-Search Problem

The spatial-search problem is defined as follows. Consider at time j=0 a fully delocalized walker on the grid, i.e., $\forall \left(p,q,a\right)\in {\left[\;\left[0,M\right]\;\right]}^{2}\times \left\{L,R\right\},\psi_{0,p,q}^{a}=\frac{1}{M\sqrt{2}}$. The problem addressed by the Coulomb walk with this initial condition is: can the walker localize on the nodes where $\phi_{p,q}$ is at its extremum, that is, the four nodes around the center $\left(\Omega_{x},\Omega_{y}\right)=\left(\frac{M}{2}-\frac{1}{2},\frac{M}{2}-\frac{1}{2}\right)$?

The first observable will be the probability of being on these nodes as a function of time and of the number of grid nodes: (12)$$P_{j}\left(N\right):=\sum_{\left(p^{\prime },q^{\prime }\right)\in {\left\{\pm \frac{1}{2}\right\}}^{2}}{∥\Psi_{j,\Omega_{x}+p^{\prime },\Omega_{y}+q^{\prime }}\left(N\right)∥}^{2}\;,$$
which we call *localization probability*. It has been shown in Ref. [29] that the localization probability admits a first maximum at time $j_{1}\left(N\right)=82$, independent of *N*. We now define long times as times $t_{j}$ with *j* much larger than 82. The long-time behavior is studied below in Section 3.

We consider the *probability distribution* over space as second observable,
(13)$$d_{j,p,q}\left(N\right):={∥\Psi_{j,p,q}\left(N\right)∥}^{2}\;,$$
which is studied in Section 6.

The fully delocalized initial condition is common in spatial-search problems because of Grover’s algorithm [20]. Moreover, this initial condition can easily be implemented on a quantum circuit as a tower of Hadamard gates. Other initial superpositions for the coin part were considered in Ref. [29]. Now, the fully delocalized initial condition forces us to pay attention to boundary conditions. In our work, we choose periodic boundary conditions. From a computer science point of view, one can expect from a database to have a list of adresses, which are on a graph whose ends are connected, corresponding exactly to periodic boundary conditions.

## 3. Noiseless Case: Long Times

In Ref. [29], it is shown that for a ‘small’ grid (up to $N=2.5\times 10^{5}$), the first maximum occurs at $j_{1}\left(N\right)=82=O_{N\to \infty }\left(1\right)$ with a localization probability $P_{j_{1}\left(N\right)}\left(N\right)$ scaling as O(1/N). According to Figure 3, the result $j_{1}\left(N\right)=O\left(1\right)$ actually holds up to $N=9\times 10^{6}≃10^{7}$. The left panel of Figure 4 shows that $P_{j_{1}\left(N\right)}\left(N\right)=O\left(1/N\right)$ is valid up to $N=900\times 900≃10^{6}$. Now, $P_{j}\left(N\right)$ with fixed *N* presents several other maxima as *j* varies, and Figure 4 shows in particular that there is a prominent second maximum. This second maximum occurs at a time $j_{2}\left(N\right)$ which, according to Figure 3 and to the right panel of Figure 4 scales as $O(\sqrt{N})$. The right panel of Figure 4 also shows the localization probability $P_{j_{2}\left(N\right)}\left(N\right)=O\left(1/\ln N\right)$. This result matches the state-of-the-art result in 2D DQW search algorithms before amplitude amplification [26].

## 4. Ressource Analysis

Since the evolution operator of the Coulomb walk is built out of two 1D shift operators, one for each spatial directions, the Coulomb walk only requires a 2-dimensional coin space. On the contrary, Tulsi’s walk (see Ref. [26]) uses a 2D shift operator, which requires a 4-dimensional Hilbert space for the coin, so encoding this walk requires one more qubit than encoding the walk studied in the present article. Also note that Tulsi’s algorithm also uses an ancilla qubit to allow a part of the probability amplitude to remain on the same site after one evolution step (ltechnically, Tulsi’s walk uses a controlled shift operator and a controlled coin operator with respect to the ancilla). Thus, in total, Tulsi’s algorithm needs two more qubits than the Coulomb walk to perform a quantum spatial search on a database of the same size. Roget et al. ’s walk, presented in Ref. [28], is a DQW—as is the Coulomb walk. It also uses two 1D shift operators and dispenses with the ancilla qubit. The difference with the Coulomb walk lies in the choice of oracle. The Coulomb walk uses an artificial electric field as oracle, while Roget et al.’s walk views the node to be found as a defect and therefore replaces on the defect the rotation R(θ) of Equation (Equation 4) by the identity operator.

A scheme implementing efficiently (up to a given precision ϵ) position-dependent diagonal unitaries similar to the electric potential oracle in Equation (Equation 6) can be found in J. Welsh et al. (Ref. [30]). The total number *n* of one-qubit and two-qubit quantum operations used in this scheme scales as $O(\frac{1}{\epsilon })$ and is actually independent of *n*. However, the implementation of the shift operators $S_{1,2}$ (see Equation (3)) requires a number of primitive quantum operations, which does depend on *n* and scales as $O(n^{2})$ because implementing shift operators requires performing Quantum Fourier Transforms (QFTs) [41]. Note that each coin operation $R(\theta^{-})$ and $R(\theta^{+})$ in Equation (2) can be implemented as only one single quantum gate on the coin qubit.

## 5. Oracle Noise

Today, one of the main goals in quantum computing is having fault-tolerant algorithms, which can be implemented on NISQ devices [42,43,44].

In the scheme developed by Welch et al. in Ref. [30], the final quantum circuit of the oracle is composed of CNOT and $R_{Z}$. The rotation angles are only implementable up to a finite accuracy due to hardware limitations. This generates fluctuations in the potential ϕ and we model these fluctuations by a white noise. More precisely, we replace $\phi_{p,q}$ by $\phi_{j,p,q}^{B}=\phi_{p,q}+B_{j,p,q}$, where *B* is a white noise in all its variables. To make things as simple as possible, given a point (j,p,q), $B_{j,p,q}$ is chosen randomly with uniform distribution in a certain interval $\left(-B_{\max},B_{\max}\right)$ independent of (j,p,q). Noise that depends on time only does not modify the probability distribution. All noises considered in this article will therefore be space-dependent. We will first focus on time-independent, but space-dependent noise, and then switch to both time- and space-dependent noise.

Note that decoherence noise on the free-walk part and on Grover search has already been studied in Refs. [45,46,47,48,49,50,51].

The amplitude of the noise is best characterized by the noise-to-signal ratio: (14)$$r:=\frac{B_{\max}}{\max_{p,q}|\phi_{p,q}|}\;.$$

### 5.1. Spatial Oracle Noise

In this subsection, all observables are averaged over 50 realizations of the noise. Figure 5 presents results obtained for $N=200^{2}$ and $N=500^{2}$. When the noise-to-signal ratio *r* is not too high, say r≲0.5, both peaks still exist and the second one occurs slightly later, with approximately the same time delay with respect to the noiseless situation. The amplitude of the peaks is also affected by the noise. In particular, for large enough *N* (see the right panel in Figure 5), the amplitude of the first peak decreases while the amplitude of the second peak actually increases. Thus, weak noise favors, and even enhances the second peak, at least for large enough values of *N*. Increasing the noise-to-signal ratio *r* erases the first peak and, to a certain extent, also the second one. Note however that, for large enough *N*, the probability $P_{j}^{r}\left(N\right)$ still exhibits a (rather flat) maximum in lieu of the second peak. So, in any case, noise favors the second peak. So, all in all, the algorithm studied in this article shows good to great robustness to spatial noise. It is also instructive to investigate this robustness through the probability distribution over space $d_{j,p,q}^{r}\left(N\right)$ and this is done in Section 6 below.

### 5.2. Spatiotemporal Oracle Noise

In this subsection, all observables are averaged over 10 realizations of the noise. Numerical results are presented in Figure 6. One first observes a global decreases in the localization probability, which gets globally lower with increasing *r*. However, it also appears that the first peak is less impacted by the noise than the rest of the curve, and especially the second peak. This can be understood in the following way. Since the noise we are considering is white in both space and time, the central limit theorem applies. The walk will therefore exhibit diffusive behavior in the ‘long’-time limit (see, for example, Ref. [49]).

However, the shorter the time, the less important the perturbation induced by the noise on the walk’s behavior. The striking robustness of the first peak, which always occur at j=82, indicates that j=82 is a ‘short’ time, at least for noise-to-signal ratios exceeding 0.5.

## 6. Probability Distribution in Space

We now investigate the probability distribution $D_{j}=\left\{d_{j,p,q},\left\{p,q\right\}\in {\left[\;\left[0;M\right]\;\right]}^{2}\right\}$ of the walk at the localization times $j_{1}$ and $j_{2}$ corresponding to the first and second peak. The *height ratio* η between the peak and the background is defined as
(15)$$\eta_{j}(N):=\frac{d_{j,\frac{M}{2}-1,\frac{M}{2}-1}\left(N\right)}{d_{j,1,1}\left(N\right)}\;,$$
where $d_{j,\frac{M}{2}-1,\frac{M}{2}-1}\left(N\right)$ is the probability to be on one of the four nodes of interest (where the potential is maximum), and where $d_{j,1,1}\left(N\right)$ is the probability to be where the potential is the weakest.

### 6.1. Noiseless Case

The noiseless case is presented in Figure 7. The probability distributions are sharply peaked on the nodes of interest for both $j=j_{1}$ (top plots) and $j=j_{2}$ (bottom plots). For a small grid size (i.e., $\sqrt{N}=200$), the height ratio is better for the first peak than for the second peak (the precise values are given in the figure caption). For a larger grid size (i.e., $\sqrt{N}=500$ and $\sqrt{N}=1000$), the height ratio of the first peak is important, but that of the second peak is substantially larger (see the figure caption).

### 6.2. Spatial Oracle Noise

Let us now investigate the probability density $D_{j}^{r}=\left\{d_{j,p,q}^{r},\left\{p,q\right\}\in \left[\;\left[0,M\;\right]\right]\right\}$ in the presence of noise with noise-to-signal ratio *r*. Figure 8 displays $D_{j}^{r}$ (top plots) and $D_{j}^{r}-D_{j}$ (bottom plots) at $j=j_{1}$ (left plots) and $j=j_{2}$ (right plots). On the top plots of Figure 8, where $D_{j}^{1/3}$ for r=1/3 is plotted, one observes that the overall shapes of the peaks, and in particular their widths, are not affected by the noise. The height ratios (given in the caption of Figure 8) are still very large, even in the presence of a substantial amount of noise (r=1/3). This shows *not only good, but high robustness* of the walk to spatial noise. Looking at the bottom plots of Figure 8 one observes that noise makes the first peak lower (two bottom left plots), but makes the second peak (two bottom right plots) higher for a small grid size (M=200) or balanced between the four nodes of interest for a larger grid size (M=500). These observations are of course consistent with the curves of Figure 5.

## 7. Conclusions and Discussion

In this paper, we have shown that the 2D electric Dirac DQW presented in Ref. [29] has at least two different localization peaks: (i) one at short times ($O_{N\to \infty }\left(1\right)$ with *N* the number of nodes on the 2D grid), for which the localization probability scales as O(1/N), and (ii) another at a time scaling as $O(\sqrt{N})$ with localization probability in O(1/lnN), which matches the state-of-the-art result in spatial search with 2D DQWs before amplitude amplification [26,28]. This dynamic was studied numerically up to $N=9\times 10^{6}≃2^{20}$.

This quantum spatial search also presents a memory advantage by formally requiring two qubits less than Tulsi’s algorithm. In terms of quantum operations, the oracle can be efficiently implemented on a quantum circuit up to an error ϵ using $O(\frac{1}{\epsilon })$ primitive quantum gates, allowing its implementation on current NISQ devices and future fault-tolerant universal quantum computers.

We have also explored the effect of oracle noise by adding a white noise to the electric potential. This white noise can be viewed, for example, as a model of the fluctuations induced by the finite accuracy implementation of the quantum rotations involved in the Oracle quantum circuit [30]. Our results demonstrate that the algorithm is highly robust to oracle noise. The second peak is not only highly robust to, but actually slightly amplified by, spatial noise. The second peak is admittedly less robust to spatiotemporal noise but the first peak turns out highly robust to this type of noise. This study is thus very encouraging for the future implementation of quantum spatial search with electric potential on universal quantum computers and NISQ devices.

Adapting to the present walk, the ancilla technique used in Tulsi’s walk may make the second peak appear sooner and might eventually help the walk reach Grover’s lower bound. Furthermore, studying the evolution of the localization probability under other kinds of noises is assuredly very promising to extend the robustness properties of the quantum spatial search with electric potential. Finally, extending all results to higher dimensions and to walks using other fields such as oracle will certainly prove interesting.

## Figures and Tables

**Figure 1 entropy-24-01778-f001:**
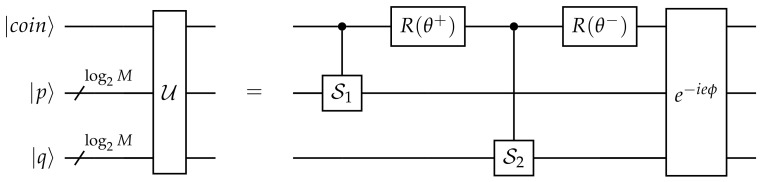
Quantum circuit of a single step operator U.

**Figure 2 entropy-24-01778-f002:**
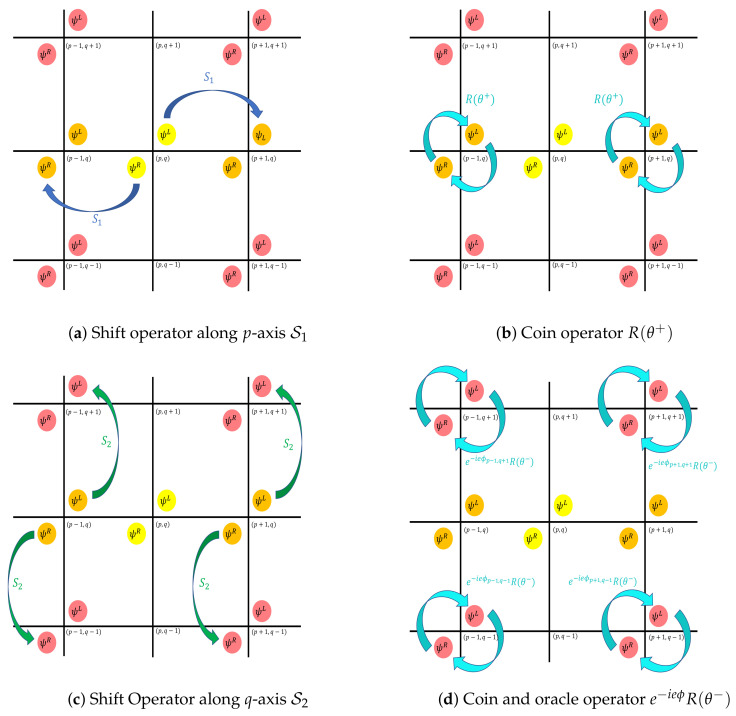
Schematic representation of the quantum walk scheme for one step, starting from position (p,q). (**a**) First, the $S_{1}$ shift operator is applied, shifting the $\psi^{L}$ component at position (p,q) to position (p+1,q) and the $\psi^{R}$ component at position (p,q) to position (p−1,q). (**b**) Second, the rotation $R(\theta^{+})$ is applied at positions (p±1,q), mixing the two components $\psi^{L}$ and $\psi^{R}$ (see Equation (Equation 4) with angle $\theta^{+}$). (**c**) Third, the $S_{2}$ shift operator is applied, shifting the $\psi^{L}$ components at positions (p±1,q) to positions (p±1,q+1) and the $\psi^{R}$ components at position (p±1,q) to position (p±1,q−1). (**d**) Finally, the rotation $R(\theta^{-})$ and the oracle $e^{-ie\phi }$ is applied. The two components $\psi^{L}$ and $\psi^{R}$ are first mixed by the rotation defined Equation (Equation 4) with angle $\theta^{-}$ and then multiplied by the position-dependent phase factor defined by the potential ϕ. In this scheme, these operations are illustrated by considering the components $\psi_{p,q}^{L}$ and $\psi_{p,q}^{R}$ at a node (p,q). At the end of one step, the components $\psi_{p,q}^{L}$ and $\psi_{p,q}^{R}$ are spread at the nodes (p+1,q+1),(p+1,q−1),(p−1,q+1),and(p−1,q−1) in a unitary manner.

**Figure 3 entropy-24-01778-f003:**
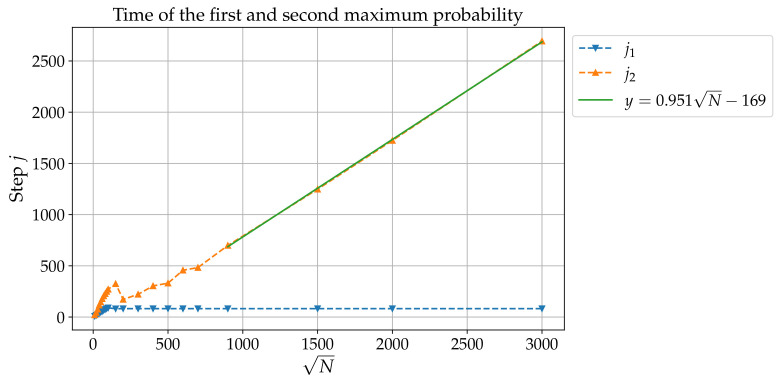
Times $j_{1}$ (blue) and $j_{2}$ (orange) at which the localization probability $P_{j}\left(N\right)$ reaches a maximum, plotted as a function of $\sqrt{N}$ for m=0, e=−1, and Q=0.9.

**Figure 4 entropy-24-01778-f004:**
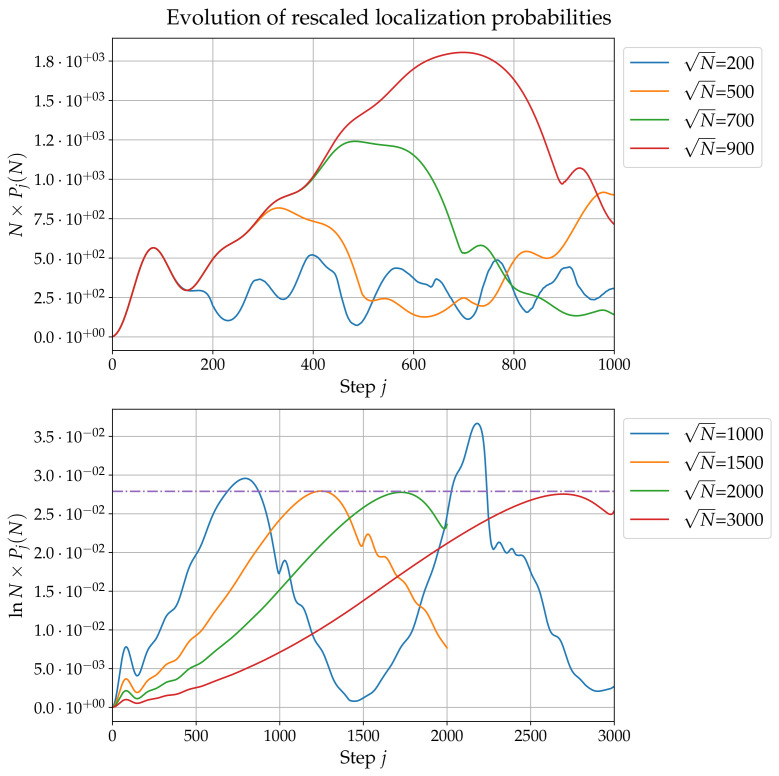
Rescaled localization probability $P_{j}\left(N\right)$ for m=0, e=−1, and Q=0.9. Left panel: $P_{j}\left(N\right)\times N$ as a function of *j*, for several values of *N*. Right panel: $P_{j}\left(N\right)\times \ln N$ as a function of *j*, for different values of *N*.

**Figure 5 entropy-24-01778-f005:**
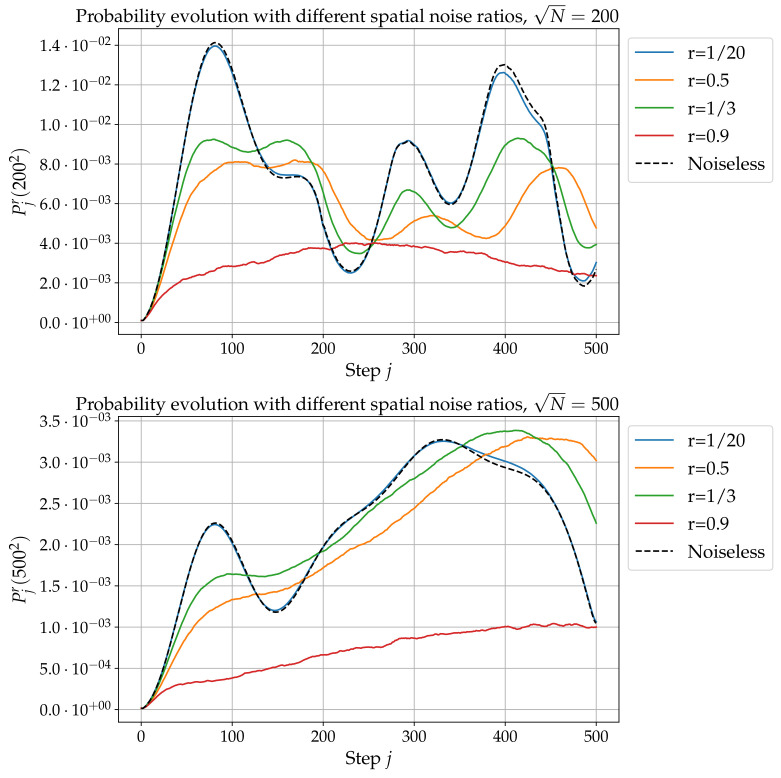
Localization probability $P_{j}^{r}\left(N\right)$ with spatial noise as a function of *j*, for different noise-to-signal ratios *r*, for m=0, e=−1, Q=0.9, and $\sqrt{N}=200$ (top) and $\sqrt{N}=500$ (bottom).

**Figure 6 entropy-24-01778-f006:**
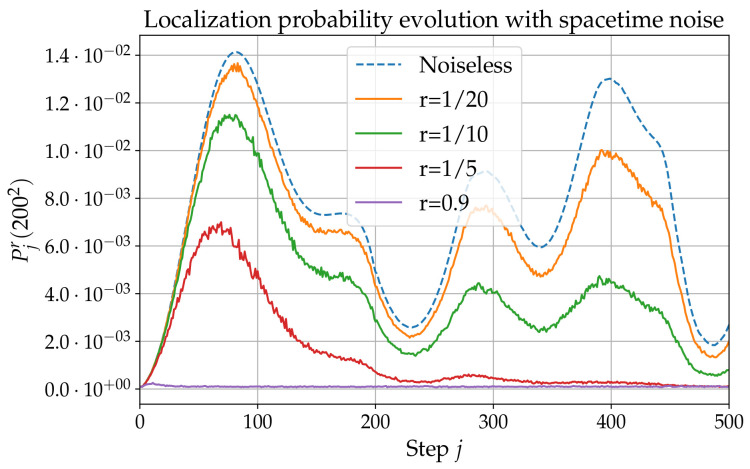
Localization probability $P_{j}^{r}\left(200^{2}\right)$ as a function of *j* for different spatiotemporal noise-to-signal ratios *r*, with m=0, e=−1, and Q=0.9.

**Figure 7 entropy-24-01778-f007:**
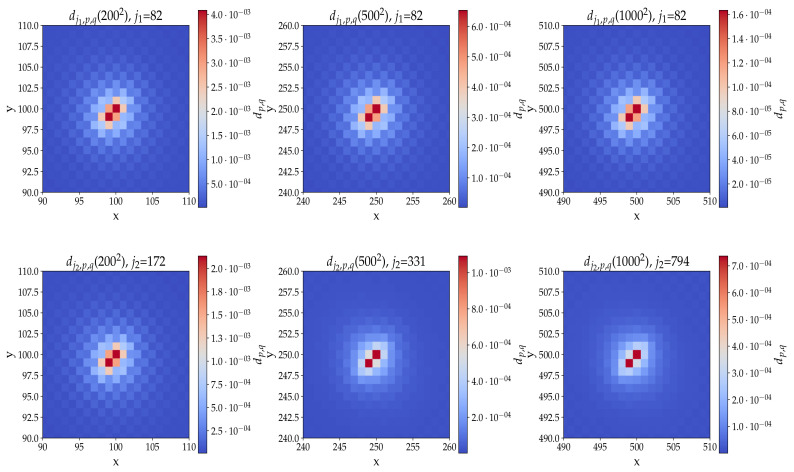
Probability distribution $D_{j}$ in the noiseless case for $\sqrt{N}=200,500,1000$, $j=j_{1}$ (**top** plots), and $j=j_{2}$ (**bottom** plots) and m=0, e=−1, Q=0.9. Height ratios for $j_{1}$: $\eta_{j_{1}}\left(200^{2}\right)=160$, $\eta_{j_{1}}\left(500^{2}\right)=163$, and $\eta_{j_{1}}\left(1000^{2}\right)=163$. Height ratios for $j_{2}$: $\eta_{j_{2}}\left(200^{2}\right)=127$, $\eta_{j_{2}}\left(500^{2}\right)=242$, and $\eta_{j_{2}}\left(1000^{2}\right)=1933$.

**Figure 8 entropy-24-01778-f008:**
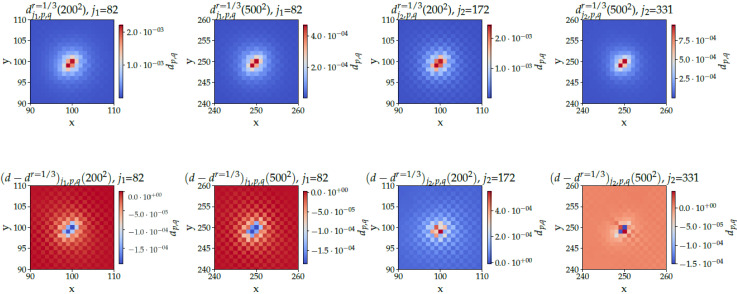
**Top** plots: Probability distribution $d_{j,p,q}^{r=1/3}\left(N\right)$ for $\sqrt{N}=200$ and 500, at $j_{1}$ (left plots) and $j_{2}$ (right plots), averaged over 50 realizations of the spatial noise, with m=0, e=−1, Q=0.9. Height ratios for $j_{1}$: $\eta_{j_{1}}^{r=1/3}\left(200^{2}\right)=87$ and $\eta_{j_{1}}^{r=1/3}\left(500^{2}\right)=115$. Height ratios for $j_{2}$: $\eta_{j_{2}}^{r=1/3}\left(200^{2}\right)=142$ and $\eta_{j_{2}}^{r=1/3}\left(500^{2}\right)=218$. **Bottom** plots: Difference $d_{j,p,q}^{r=1/3}\left(N\right)-d_{j,p,q}\left(N\right)$ between the noisy and the noiseless cases. for m=0, e=−1, Q=0.9.

## Data Availability

Data are available on reasonable demand at fredonthibault@gmail.com.

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
