# Peer review of "Quantum Spatial Search with Electric Potential: Long-Time Dynamics and Robustness to Noise"

_entropy, 2022, doi:10.3390/e24121778_

Round 1

Reviewer 1 Report

The study of long-time dynamics and robustness to noise for DQW problems seems to be an interesting exercise. I would recommend this for publication.

This paper extends the analysis of their Ref 1, where discrete quantum walk for massless Dirac fermions on a 2d lattice with Coulomb field living on the lattice sites was studied. The previous reference noticed that a marked node corresponds to a peak in the localization probability in O(1) time steps. The extension of their analysis for longer time scales has been carried out in this paper and leads the authors to conclude that there is a second peak. Also, an analysis of the robustness to noise has been carried out. The paper provides a detailed numerical analysis to justify their findings and their plots are illuminating.

Author Response

We thank the editor and the reviewer for their expert work on our manuscript. We are happy that the reviewer thinks our manuscript can be published as is in entropy.

Reviewer 2 Report

In this paper, the authors have presented various results of a spatial search model based on electric Dirac quantum walk - published previously as Entropy 2021, 23(11), 1441. The two observable quantities that are relevant to spatial search problem are - localization probability and probability distribution over space. Here, the authors discusses two important aspects of their model: (i) large time behavior and (ii) Effect of noise (spatial and spatiotemporal). The long time behavior reveals an emergence of a second peak in the localization probability. The noise enters into the picture in the implementation of the quantum circuit of the oracle. This noise is modeled as white noise with uniform distribution and is characterized by noise to signal ratio. For spatial oracle noise, the peaks are highly affected by the noise to signal ratio and it diminishes at higher values of the ratio. With spatiotemporal noise, the authors report a global decrease in the localization probability. The probability distributions is not affected by the spatial oracle noise. The work presented in this paper requires some revisions:

1. A schematic diagram of the system will be very helpful for better readability of the paper. 2. In Figure 2, the plot legends doesn’t match the figure caption.

Author Response

We thank the editor and the reviewer for their expert work our manuscript. We have carried out the two suggestions made by the reviewer. We have added two figures which offer schematic presentation of the algorithm. We have also fixed the scaling law figure caption. We thank again the reviewer for the helpful suggestions.